# Degradation of Bacterial Antibiotic Resistance Genes during Exposure to Non-Thermal Atmospheric Pressure Plasma

**DOI:** 10.3390/antibiotics11060747

**Published:** 2022-05-31

**Authors:** Ibtissam Courti, Cristina Muja, Thomas Maho, Florent P. Sainct, Philippe Guillot

**Affiliations:** Laboratoire Diagnostics des Plasmas Hors-Equilibre (DPHE-EA 4562), University of Toulouse, I.N.U. Champollion, 81000 Albi, France; ibtissam.courti@univ-jfc.fr (I.C.); thomas.maho@univ-jfc.fr (T.M.); florent.sainct@univ-jfc.fr (F.P.S.); philippe.guillot@univ-jfc.fr (P.G.)

**Keywords:** wastewater treatment technologies, plasma, ARB, ARGs, HGT

## Abstract

Bacterial resistance to antibiotics has become a major public health problem in recent years. The occurrence of antibiotics in the environment, especially in wastewater treatment plants, has contributed to the development of antibiotic-resistant bacteria (ARB) and the spread of antibiotic resistance genes (ARGs). Despite the potential of some conventional processes used in wastewater treatment plants, the removal of ARB and ARGs remains a challenge that requires further research and development of new technologies to avoid the release of emerging contaminants into aquatic environments. Non-thermal atmospheric pressure plasmas (NTAPPs) have gained a significant amount of interest for wastewater treatment due to their oxidizing potential. They have shown their effectiveness in the inactivation of a wide range of bacteria in several fields. In this review, we discuss the application of NTAPPs for the degradation of antibiotic resistance genes in wastewater treatment.

## 1. Introduction

Antibiotics are one of the most important discoveries in human history. They have proven to be effective in the treatment and control of human infections and mortality [1]. They have also become fundamental in agriculture and veterinary medicine. For several decades, global consumption of antibiotics has increased due to population growth and demand. Extensive use of antibiotics has led to the emergence of antibiotic-resistant bacteria (ARB). Antibiotic resistance is the ability of bacteria to resist the action of an antibiotic to which they were previously susceptible. Bacteria acquire resistance to antibiotics by chromosomal DNA mutations that alter the existing bacteria proteins or by horizontal gene transfer (HGT) which allows the acquisition of new genetic material between bacteria that are not in a parent–offspring relationship [1]. Antibiotic resistance constitutes a critical worldwide risk for humans, animals, and the environment. In 2014, Lord Jim O’Neill and his team estimated that antibiotic-resistant bacteria could cause 10 million deaths per year by 2050 [2]. In 2017, the WHO announced that antibiotic-resistant bacteria pose the highest risk to human health. Antimicrobial resistance causes 25,000 deaths in Europe and 23,000 deaths in the United States annually [3].

Aquatic environments, such as wastewater treatment plants (WWTPs), are considered important reservoirs and pathways for the dissemination of antibiotic-resistant bacteria (ARB) and antibiotic resistance genes (ARGs) [4,5,6,7]. Several studies have highlighted the direct relationship between human activity and the spread of bacteria and antibiotic resistance genes in aquatic environments [4]. Various conventional treatment methods have been developed to eliminate ARB and degrade ARGs from wastewater, including membrane filtration [8], chlorination [9], ozonation [10], and ultraviolet irradiation (UV) [11]. Sharma et al. reviewed the impact and the mechanisms of these treatments on the inactivation of ARB and the degradation of ARGs in wastewater [12]. However, it has been reported that these treatment methods are not able to effectively reduce the level of ARB and ARGs, which are released into aquatic environments, such as lakes and rivers [4,13]. For instance, the abundance of antibiotic-resistant bacteria on the membrane surface during the membrane filtration process creates an environment that favors horizontal gene transfer, resulting in the production of more ARB and ARGs [8]. Macauley et al. compared the inactivating effects of chlorine, UV, and ozone decontamination processes on aureomycin-, sulfamethazine-, clindamycin-, and tetracycline-resistant bacteria [14]. The results showed that chlorine decontamination was the most effective, followed by UV and ozone. However, they reported that the frequency of ARG transfer increased after UV and chlorine decontamination. Similarly, Stange et al. reported that UV and chlorine treatments were effective in inactivating bacterial cells, but incomplete degradation of ARGs was observed [15]. Conversely, other studies indicated that with high doses, chlorine can efficiently degrade ARGs. Zhuang et al. investigated the inactivation of *sul1* and *tetG* and showed that exposure to 40 mg/L of chlorine for 60 min resulted in a 1.65–2.28-log reduction in these genes. However, the chlorine concentration needed to obtain these results seems too high to be considered as a treatment solution [16].

Recently, advanced oxidation processes (AOPs) based on free radical oxidation such as UV/H_2_O_2_ [17], Fenton oxidation [18], photocatalytic oxidation [19], and nanomaterials [20], have received considerable attention for the removal of ARB and ARGs in wastewater. Even though these methods were able to decrease ARB and ARG levels, the exposure of ARB to sub-lethal doses of some of these decontamination processes could allow recovery from injury and enhance the frequency of horizontal gene transfer. Several studies reported that gene transfer between bacteria could be induced by environmental factors such as organic contaminants [21], antibiotics [22], biocides [23], and oxygen species produced during advanced oxidation processes [24,25]. Further research suggests that environmental stressors enhance the horizontal transfer of ARGs between bacteria, which occurs through generally conserved cellular response pathways, including those involved in ROS response systems. The involvement of these cellular pathways and mechanisms poses important questions about the possible role of other environmental factors in ARG transmission [25].

Non-thermal atmospheric pressure plasma (NTAPP), one of the advanced oxidation processes, has recently attracted a significant amount of interest for wastewater decontamination. Plasma is a reactive cocktail of various reactive oxygen and nitrogen species, such as hydrogen peroxides, hydroxyl radicals, ozone, nitric oxide, UV photons, and charged particles such as electrons and ions, which are responsible for the inactivation of microorganisms [26]. Several pilot-scale studies focusing on the decontamination capacities of plasmas have been carried out with samples from municipal wastewater treatment plants (WWTPs) [27,28,29,30,31,32]. A large and growing body of literature reports the effective inactivation capacity of NTAPP for a wide range of microorganisms. The effect of plasma has been evaluated on antibiotic-resistant bacteria, antibiotic resistance genes, and the mechanisms of horizontal transfer of resistance genes. This research will provide an overview of recent advances in the application of NTAPP for the removal of ARB and ARGs from wastewater.

## 2. Antibiotic-Resistant Bacteria and Antibiotic Resistance Genes in WWTPs

There are several routes for the dispersal of ARGs in aquatic environments. One of them is vertical gene transfer (VGT), a mechanism that allows the transmission of modified genetic information to subsequent generations. The second path of dispersal is horizontal gene transfer (HGT) between microorganisms that do not have a parent–offspring relationship. This process is mediated by mobile genetic elements (MGEs) such as plasmids, transposons, integrons, and bacteriophages. Horizontal gene transfer is considered to be a non-reproductive gene transfer that consists of the exchange of genetic information between different bacterial species [33]. HGT is the most involved mechanism for the dissemination of resistance in WWTPs. There are four different mechanisms of HGT (Figure 1) [34]. Conjugation is the transfer of DNA from a donor bacterium to a recipient bacterium, which involves physical contact between the two bacteria via sexual pili. Conjugation is facilitated by the presence of a fertility factor (F^+^) that allows the donor bacterium to develop its conjugation machinery for gene transfer. Transformation involves the integration of a naked fragment of extracellular DNA into the cytoplasm or chromosome, which can be received by a competent bacterium. Transduction is a process that consists of the transfer of DNA from a donor bacterium to a recipient bacterium, through a viral vector (a bacteriophage). Finally, gene transfer agents (GTAs) are bacteriophage-like particles that carry random pieces of the producer cell’s genome. GTA particles can be released by cell lysis and spread into a recipient cell.

There is a large body of research that points out the presence of ARB in WWTPs and their persistence after wastewater treatment. Table 1 shows an overview of ARB and ARGs identified in wastewater effluents. For example, Verburg et al. examined the fate of ARB after wastewater treatment in a municipal WWTP in the Netherlands. They analyzed 2886 isolates of three different species (*Escherichia coli, Klebsiella* spp., *Aeromonas* spp.) obtained before and after wastewater treatment and showed that even if the bacterial concentration was reduced by around 2 log, the proportion of antimicrobial-resistant bacteria did not decrease after wastewater treatment [35]. Luczkiewicz et al. evaluated the abundance of enterococci in a WWTP [36]. The predominant species were *Enterococcus faecium* (60.8%) and *Enterococcus faecalis* (22.1%), where 90% showed resistance to at least one antibiotic. Notably, bacteria showed resistance to nitrofurantoin and erythromycin (53% and 44%, respectively), and also to ciprofloxacin (29%) and tetracycline (20%). The abundance of these resistant species in the WWTP was above 4 log/100 mL. Neudorf et al. demonstrated the occurrence of bacteria harboring resistance to several classes of clinically pertinent antibiotics at three wastewater treatment plants in Northern Canada [37]. Guo et al. investigated the diversity of ARGs and MGEs in a WWTP [38]. Metagenomic analysis revealed that activated sludge and digested sludge showed different microbial communities and changes in the types and occurrence of ARGs and MGEs. A total of 42 ARG subtypes were identified in the activated sludge, while 51 ARG subtypes were detected in the digested sludge. In addition, MGEs including plasmids, transposons, integrons (*intI1*), and insertion sequences were abundant in both sludge samples.

Several studies showed that hospital wastewater had high concentrations of ARB and ARGs compared to municipal wastewater. Grabow and Prozesky were the first to report the abundance of antibiotic-resistant coliforms of hospital origin in sewage treatment plants [48]. Rowe et al. investigated the abundance of ARGs in wastewater samples from three locations in Cambridge [49]. They found that the ARG abundance in hospital effluent was 9-fold higher than in agricultural effluent and 34-fold higher than in river source water. Gaşpar et al. compared the antibiotic resistance levels of *Escherichia coli* in samples collected from two hospital wastewaters and two municipal wastewaters [50]. Antibiotic susceptibility testing was performed on 81 *E. coli* isolates (47 from hospital wastewater and 34 from municipal wastewater). Multidrug resistance was observed in 85.11% of hospital wastewater isolates and 73.53% of municipal wastewater isolates. A comparative study of the abundance of antibiotic resistance genes in the effluent of three hospitals in Romania showed the presence of genes encoding resistance to tetracyclines, aminoglycosides, chloramphenicol, β-lactams, sulfonamides, quaternary ammoniums, streptogramin B antibiotics, and macrolidelincosamide with high concentration levels between 0.01 and 0.1 copies per 16S rRNA gene copy measured by qPCR [51]. A recent overview by Hassoun-Kheir et al. reported that 81% of the studies showed that hospital wastewater contains higher amounts of ARB and ARGs than municipal wastewater [52].

In addition, hospital wastewater also contains relatively high amounts of antibiotics. Sabri et al. reported that most antibiotics are not completely metabolized in humans and animals, meaning that 40–90% of the active substance is excreted and thus ends up in wastewater [53]. High concentrations of a wide range of antibiotics were reported in hospital and urban wastewater, including β-lactams, ciprofloxacin, sulfamethoxazole, quinolones/fluoroquinolones, sulfonamides, and tetracyclines [54]. For example, fluoroquinolones were detected at the highest concentration, especially in hospital effluent samples, and ciprofloxacin and sulfamethoxazole showed almost 10-fold higher concentrations downstream than upstream of the WWTP discharge. A number of studies have reported that continuous exposure of bacteria to sub-inhibitory concentrations of antibiotics exerts selective pressure and promotes the spread of ARB and ARGs in wastewater treatment plants.

Despite the current processes used in wastewater treatment plants to reduce ARB, ARG, and MGE levels, they are still detected in the environment and greatly participate in the spread of antibiotic resistance. Furthermore, as mentioned above, the accumulation of antibiotics imposes a high selective pressure in the environment, which facilitates the acquisition of resistance mechanisms by bacteria. In this context, the inefficiency of wastewater treatment processes confirms the necessity to develop new, more efficient and accessible wastewater treatment technologies that can remove emerging contaminants, in particular ARB and ARGs, and thus prevent their spread in aquatic environments.

## 3. Plasma Discharge and Chemistry

Plasma is a partially or completely ionized gas, composed of electrons, free radicals, ions, and neutrals. It can be generated by applying thermal energy, an electric field, or electromagnetic radiation energy to a gas. However, attention has been focused mainly on plasmas generated by electric fields [55,56,57,58]. An applied electric field can transfer energy to any free electrons present in a gas. These high-energy electrons transfer their energy to neutral species in the gas by collisions, thus generating an ionized gas. Depending on the thermal equilibrium between the electrons (Te) and gas (Tg), plasmas can be divided into non-thermal plasmas and thermal plasmas. Thermal plasmas are in thermodynamic equilibrium (Te = Tg), whereas non-thermal plasmas are out of thermodynamic equilibrium (Te >> Tg). Non-thermal plasma can be generated at normal atmospheric pressure.

Non-thermal atmospheric pressure plasmas (NTAPPs) have attracted considerable interest for decades because of their rich chemistry [56,59,60,61]. When a non-thermal plasma is generated, the energy of the electric field accelerates free electrons and ionizes the atoms and molecules of the gas, releasing various reactive species including oxygen species (ROS) and nitrogen species (RNS). These species are called the primary reactive species. They are characterized by very short lifetimes; for example, the lifetimes of OH, NO, and O_2_^−^ * radicals are 2.7 µs, 1.4 µs, and 1.3 µs, respectively [56]. Because of their high reactivity, some of these primary reactive species react immediately with the surrounding gas to create secondary reactive species. The reactive oxygen and nitrogen species (ROS and RNS) produced in the plasma phase and the plasma–liquid interface reach the target, such as a liquid, and are dissolved, forming tertiary reactive species including H_2_O_2_, O_3_, NO_2_, and NO_3_. These reactive species have very long lifetimes, from a few milliseconds to several days. Upon contact with a liquid, more reactive species are produced such as ONOO^−^, ONOOH, NO_2_^−^, NO_3_^−^, and H_2_O_2_, resulting in a decrease in the pH of the target liquid of up to 2 pH values. Figure 2 shows a schematic of the formation of ROS and RNS in the discharge region, the plasma phase, the plasma–liquid interface, and inside the liquid target [61].

Several plasma sources, including corona discharges, dielectric barrier discharges (DBD), and contact glow discharges, have been investigated for wastewater treatment [60,61,62,63]. They operate with different power signals (continuous wave in a wide range of frequencies, pulsed voltage signals), different electrode configurations such as plate-on-plate, plate-on-pin, or pin-on-pin, and different working gases and their mixtures such as air, helium, oxygen, and argon. In general, discharges in and on liquids can be divided into three categories: discharge above the liquid surface, discharge into the liquid, and discharge into bubbles. They have been studied by many research groups to investigate the plasma–liquid interface phenomenon. Typical electrode configurations for the three different types of discharges into and onto liquids are shown in Figure 3.

Plasma discharges above the liquid are considered as gaseous discharges. They are generally more energy-efficient for water treatment due to the number of chemical reactions produced in the gas phase, which are then transferred to the liquid phase [62,63]. The properties of these discharges can be different. The configuration of the high-voltage electrode influences the energy yield of the discharge as well as the production of reactive species in the liquid phase. Moreover, it has been found that the energy efficiency of this type of plasma discharge improves by decreasing the distance between the electrode and the water surface [64]. It has been reported that the distance between the electrodes in the plasma reactor can determine the amount of ROS formed, which affects the degradation efficiency [65]. An experiment was conducted using different electrode distances from 10 to 30 mm, and an applied voltage of 30 kV, for a reaction time of 15 min. Glass is often used as a dielectric barrier, particularly quartz glass, although other types of glass have also been used. These discharges can be driven by direct, pulsed, or alternating current excitation. The configurations used to generate such discharges typically consist of a metal pin–plate configuration (Figure 3a).

Plasma discharges in a liquid, also called electrohydraulic discharges, have been studied for many years because of their importance in electrical transmission processes and their potential for water treatment [62,63]. They are interesting for water treatment because of the relatively high ratio of the contact area between the plasma and the liquid. In addition, the plasma residence time in the liquid is very short due to the exchange of high-energy electrons with the surrounding liquid [59]. Plasma discharges in a liquid are considered chemically rich discharges due to the formation of reactive species directly in the liquid which immediately interact with molecules present in the liquid. The direct electric discharge applied in the liquid generates high-temperature plasma channels which induce cavitation (vapor-filled cavities in the liquid, followed by a shock wave), superficial water oxidation, and the formation of short-lived radicals by ultraviolet photolysis [66]. They can also be generated by direct, pulsed, or alternating current excitation. The most commonly reported types of electrohydraulic discharge are pulsed arc discharge and pulsed corona discharge. A typical discharge liquid is shown in Figure 3b, in a pin-to-plate configuration.

Discharges in bubbles and cavities are considered a separate group, as they are completely surrounded by the liquid that serves as an electrode [59]. These discharges consist of bubbling gas through the discharge area. This will promote mixing of the liquid treatment, an increase in the contact area between the plasma and the liquid, and the production of reactive species. It is obvious that in such discharges, the feed gas plays a decisive role due to the absence of interaction between the plasma discharge and the ambient air. Yasuoka et al. measured the size of bubbles generated in the liquid using two feed gases, namely, argon (Ar) and helium (He) [67]. The Ar bubble discharge exhibited a large contact area with the liquid compared to the helium discharge. The material, shape, size, and orientation of the nozzle determine the shape of the bubble during its formation and its position after detachment. These characteristics greatly influence the chemistry that occurs during bubble formation, the electric field involved, and, consequently, the plasma properties. A typical method involves the pumping of gas up through a nozzle anode, placed under a grounded electrode, as shown in Figure 3c. The nozzle electrode is often placed inside a dielectric tube up to its tip to prevent the energy from leaking into the water. Many variations can be found in the literature, such as a pin anode inside a hole in a dielectric plate and different orientations of the nozzle [62].

## 4. Plasma as New Wastewater Treatment Process

Several studies have shown the effectiveness of plasma for the inactivation of bacteria, both in vegetative and spore form [28,68,69,70,71]. However, the antibiotic resistance genes released by bacteria after cell membrane alteration can be transmitted to other bacteria through the different pathways mentioned above. The elimination of ARGs is a greater challenge than the inactivation of bacteria usually discussed in the literature. Antibiotic resistance and the control of ARGs must be taken into consideration.

Table 2 presents an overview of the different plasma discharges applied for the degradation of antibiotic-resistant bacteria and antibiotic resistance genes. The first study on the inactivation of antibiotic resistance genes by non-thermal plasma at atmospheric pressure was conducted 4 years ago [27]. In this study, a plasma discharge over a liquid was applied for the inactivation of methicillin-resistant *Staphylococcus aureus* (MRSA) and its methicillin resistance gene (*mecA*). The experiments were carried out in three liquid matrices (PBS, and two models of dairy and meat wastewater). The study focused on the effect of plasma on the intracellular (i-) and extracellular (e-) *mecA* gene. Intracellular ARGs are located within the cytoplasm of bacterial cells. When the cell envelopes are disrupted, intracellular ARGs are released into the environment and become extracellular ARGs, which can be acquired by other bacteria through HGT. The results of MRSA inactivation showed that increasing the plasma intensity from 0 to 0.12 kJ/cm^2^ resulted in a 5-log bacterial reduction. For the degradation of ARGs, e-*mecA* showed a higher sensitivity to plasma treatment as compared to i-*mecA*. The degradation kinetics of i-and e-*mecA* in phosphate-buffered solutions showed that the reduction in i-*mecA* genes was significantly slower than that in e-*mecA* genes. For example, 1-log degradation of i-*mecA* required a plasma treatment of more than 0.53 kJ/cm^2^, whereas only 0.12 kJ/cm^2^ could induce >1-log degradation of e-*mecA*. Additionally, the degradation of i-*mecA* (2.49 cm^2^/kJ for dairy effluent and 2.87 cm^2^/kJ for meat effluent) was much slower than that of e-*mecA* (3.35 cm^2^/kJ for dairy effluent and 3.58 cm^2^/kJ for meat effluent). The low degradation of i-*mecA* can be explained by the protective effect of the outer cell envelope, or by the intracellular structures. In addition, organic materials in dairy and meat effluents, such as proteins and lipids, act as scavengers to quench plasma-generated radicals, in particular ROS. As a result, MRSA cells and the resistant *mecA* gene are further protected from attack by plasma reactive species. The characteristics of the liquid matrices influence the efficiency of plasma decontamination and can protect ARB and ARGs, resulting in a lower degradation efficiency.

Liao et al. employed a plasma discharge over a wastewater system for the degradation of *E. coli* harboring a plasmid (pBR322) encoding for ampicillin (*bla_TEM_)* and tetracycline (*tet*) resistance [29]. During the treatment of wastewater effluents, significant reductions were observed as a function of the treatment time. The results revealed that plasma treatment at a low intensity (0.71 kJ/cm^2^) resulted in a reduction of higher than 3 log CFU/mL in *E. coli*. At a plasma intensity of 0.18 kJ/cm^2^, the degradation kinetics of e-*bla_TEM_* and e-*tet* did not decrease at all. However, when the plasma dose exceeded 0.18 kJ/cm^2^, the concentrations of e-*bla_TEM_* and e-*tet* began to decrease rapidly. The rate of degradation of the e-*bla_TEM_* and e-*tet* genes did not show significant differences (*p* > 0.05) over the range of plasma doses applied (0–20.8 kJ). In the case of the degradation of i-*bla_TEM_* and i-*tet,* the results showed that a plasma dose of 1.41 kJ/cm^2^ (8 min treatment) was sufficient to cause a reduction of more than 1 log copies/mL. This suggests that the mechanisms of degradation by plasma exposure may be the same for both resistance genes. Plasma-treatment-induced plasmid degradation may be due to plasma-generated atomic oxygen, which may react with purine and pyrimidine bases or the deoxyribose backbone of DNA.

Integrons are a key group of ARG transmission vectors in bacteria. They are genetically mobile bacterial recombination systems, called gene cassettes (GCs), that allow the acquisition and expression of sequences coding for antibiotic resistance. Several studies have reported their presence in receiving rivers after discharge from treatment plants, and their elimination during water treatment processes remains a challenge. Recently, Song et al. assessed the potential of atmospheric pressure plasmas to inactivate ARB and to destroy ARGs using a plasma bubble discharge and antibiotic-resistant *Escherichia coli* (AR *E. coli*) as a model ARB [30]. The results showed that a higher plasma voltage increased the inactivation of AR *E. coli.* Approximately 6.3 log AR *E. coli* were decreased at 18 kV within 10 minutes of treatment, whereas they were only decreased by 4.4 log AR *E. coli* at 10 kV. The production of reactive oxygen and nitrogen species (RONS) was positively related to plasma voltage, and higher voltage favoured bacterial inactivation. The study also showed that bacterial inactivation increased with increasing gas flow. The efficiency of AR *E. coli* inactivation increased with the air flow rate. Only 5.5 log AR *E. coli* were reduced in 10 minutes of plasma treatment at a flow rate of 1.5 L min-1, while up to 7.0 log AR *E. coli* were reduced when the flow rate was 2.5 L min-1. The air flow could be the source of RONS generation, in the different phases of plasma-liquid interactions, and thus the inactivation of AR *E. coli*. The minimum inhibitory concentration (MIC) representing antibiotic resistance profiles decreased by 96.9%, 96.9%, and 98.4% for the tetracycline, amoxicillin, and gentamicin tested, respectively. The production of ROS correlated positively with the plasma voltage, and a higher voltage enhanced bacterial inactivation. All ARGs decreased after the plasma treatment, with a removal efficiency of higher than 90%. The ARGs including *tetC, tetW, bla_TEM-1_, aac(3)-II*, and the integron gene *intI1* decreased by 1.04, 0.61, 1.84, 2.2, and 2.3 log copies within 10 min, respectively. The data also revealed that the plasma reduced the frequency of conjugative transfer of ARGs by approximately 63% after 10 min of treatment, thus further confirming the inhibition of the HGT of ARGs by the plasma.

Yang et al. studied AR bacteria degradation potential using a discharge in a liquid. The research group exposed *E. coli* with resistance genes (*tetA*, *tetR*, *aphA*) and the transposase gene (*tnpA*) to plasma in a 0.9% sterile saline solution [28]. The results showed that the plasma generated in the liquid was able to inactivate AR *E. coli*, eliminate ARGs, and reduce the potential of gene transfer. *E. coli* levels determined by 16S rRNA decreased by approximately 4.7 log after 15 min of discharge treatment. The gene reduction in *tetA, tetR, aphA,* and *tnpA* was increased to 5.8, 5.4, 5.3, and 5.5 log after 30 min of discharge treatment, respectively.

Overall, non-thermal atmospheric pressure plasmas present a promising alternative for the elimination of antibiotic resistance genes, as well as the mobile genetic elements involved in the spread of antibiotic resistance in wastewater treatment plants and in aquatic environments in general. A comparison of the inactivation of ARB and their associated ARGs showed the great potential of plasma processes compared to the classical advanced oxidation processes (AOPs) including electrochemical, photo-Fenton/LED, UV irradiation, and photocatalysis processes (Table 3). Plasma treatments showed relatively high inactivation efficiencies for short exposure times compared to the other AOPs. Nevertheless, full-scale studies are needed to confirm the effectiveness of plasma treatments in comparison to the different AOPs applied in wastewater treatment plants. In addition, a wide range of AOPs are frequently combined with exogenous chemicals to increase the oxidative power for better inactivation. However, the release of by-products of AOPs into the aquatic environment could pose a health risk to aquatic systems.

On the other hand, plasma at atmospheric pressure has also been studied for the degradation of antibiotics in wastewater. The decomposition of pentoxifylline in an aqueous solution was studied using an NTAPP, operating in a pulsed regime. After 60 min of plasma treatment, 92.5% of pentoxifylline was removed [76]. The degradation of veterinary antibiotics in wastewater was investigated by Kim et al. [77]. The results indicated that antibiotics were easily degraded by the plasma, and that the degradation rates were mainly governed by the amount of energy supplied. Antibiotic degradation decreased exponentially with increasing supplied energy. Based on an initial concentration of 5 mg/L, at 60% degradation efficiency, the energy requirements ranged from 0.26 to 1.49 kJ/mg, depending on the type of antibiotic substance, while those at 90% degradation efficiency ranged from 0.39 to 2.06 kJ/mg. More recently, Nguyen et al. studied the degradation of antibiotics in hospital effluents using atmospheric pressure plasma [78]. In this study, four antibiotics, namely, ofloxacin, ciprofloxacin, cefuroxime, and amoxicillin, were monitored in wastewater from seven hospitals in Ho Chi Minh City, Vietnam. Ciprofloxacin, cefuroxime, and ammonia were almost eliminated, while ofloxacin and amoxicillin were reduced by more than 72% after 15 min of plasma treatment with an applied voltage of 30 kV.

## 5. Conclusions and Future Perspectives

The limited efficiency of conventional and advanced treatment processes in wastewater treatment plants for the removal of antibiotics, ARB, and ARGs increases the demand for the development of new methods of wastewater treatment. In recent years, several studies have highlighted the advantages of applying atmospheric pressure plasmas in the microbiological decontamination of liquids. Several types of plasma discharges have been extensively studied, and comparative research on the effectiveness of different configurations on different liquid volumes and matrices has also been conducted. These studies have shown the effectiveness of plasmas for the degradation of contaminants (molecules and microorganisms) in liquids.

Future progress in this field involves an expansion toward industrial applications. Most published reports on plasma water treatment are based on lab-scale set-ups. The application of plasmas on a large scale requires a significant scientific and financial investment. In contrast to the various wastewater treatment processes used in wastewater treatment plants, plasma involves more complex and costly equipment. Plasma generation requires engineering studies, including the choice of different plasma equipment, power source design, frequency, and gas. Another problem which limits the implementation of plasma application on a large scale is that most of the literature studies single-component treatments, whether microorganisms or chemical compounds such as antibiotics, but the use of treatments with several compounds in combination is often not studied. The effectiveness of plasma against the compounds under consideration may not be reproducible under real conditions when applied to mixtures of organic materials, and interactions of reactive species between different contaminants may occur. Further studies are needed to better understand the influence of the liquid matrices and the presence of organic matter on the effectiveness of plasma treatments.

Nevertheless, the research reports included here highlight the potential of using plasma for the degradation of ARB and ARGs. These results are of particular interest in the context of the dissemination of ARB and ARGs. However, further research is needed to optimize plasma parameters for wastewater treatment and to enable its upscaling.

## Figures and Tables

**Figure 1 antibiotics-11-00747-f001:**
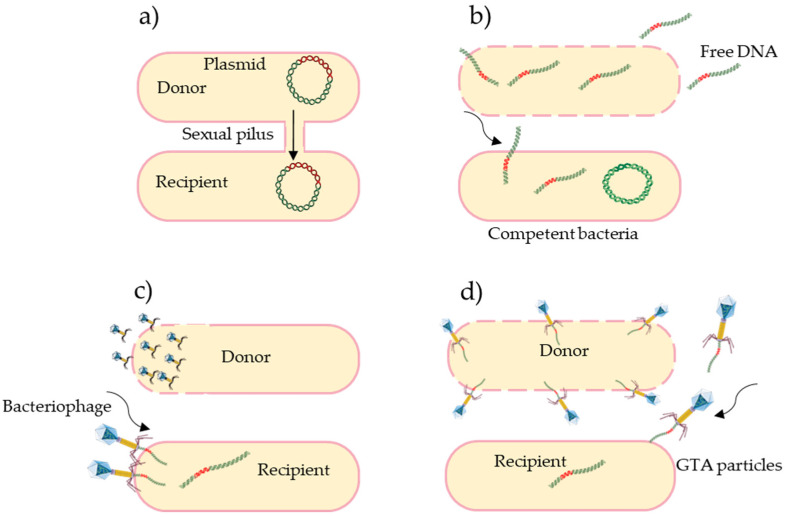
Antibiotic resistance transport mechanisms; (**a**) Conjugation, (**b**) Transformation, (**c**) Transduction, and (**d**) Gene transfer agents (GTAs), (adapted from [34]).

**Figure 2 antibiotics-11-00747-f002:**
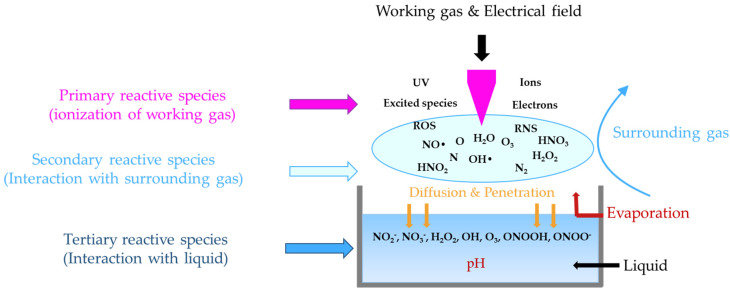
Schematic diagram of the synthesis mechanisms for plasma-liquid interaction (adapted from [61]).

**Figure 3 antibiotics-11-00747-f003:**
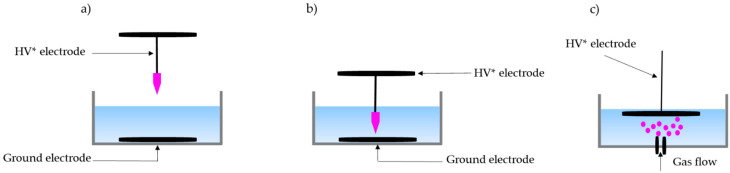
Schematic of different plasma discharge systems used for water treatment; (**a**) Discharge above liquid, (**b**) Discharge in liquid, and (**c**) Discharge in bubbles, * HV—high voltage. (adapted from [59]).

**Table 1 antibiotics-11-00747-t001:** Overview of ARB and ARGs found in the influent, effluent, and activated sludge of wastewater treatment plants (WWTPs).

ARB or ARGs	Antibiotic Resistance Profiles	Origin	WWTP Sample				Country	References
			Influent	Activated Sludge	Effluent	Effluent Concentrations of ARB or ARGs		
ARB								
*Escherichia coli*	Ciprofloxacin, cotrimoxazole, ampicillin	Nursing home, hospital, and community wastewater collection point	+	na *	+	5 log	The Netherlands	[35]
*Enterococcus*	Penicillin G, ampicillin, vancomycin	Industrial, hospital, and nursing home	+	na *	+	2 log	Germany	[39]
*Staphylococcus aureus*	Multi-resistant	Community wastewater collection point	na *	na *	+	nd *	USA	[40]
*Klebsiella* spp.	Ciprofloxacin, cotrimoxazole, ampicillin, trimethoprim	Nursing home, hospital, and community wastewater collection point	+	na *	+	5 log	The Netherlands	[35]
*Aeromonas* spp.	Ciprofloxacin, cotrimoxazole, ampicillin, trimethoprim	Nursing home, hospital, and community wastewater collection point	+	na *	+	5 log	The Netherlands	[35]
ARGs (adapted from [41])								
*ampR*	Beta-lactams	Community wastewater collection point	+	na *	+	reduction	Canada	[42]
*bla_AmpC_*		Hospital, community wastewater collection point, and receiving rivers	+	na *	+	increase	Germany	[43]
*bla_TEM_*		Hospital, domestic, and industrial	+	na *	+	increase	Portugal	[44]
*mecA*		Community wastewater collection point and receiving rivers	+	na *	+	nd *	Canada	[37]
*tetA*	Tetracycline	Community wastewater collection point	na *	+	+	nd *	Germany	[45]
		Community wastewater collection point	+	na *	+	increase	Canada	[42]
		Community wastewater collection point	+	+	+	reduction	China	[46]
		Sewage treatment plants (STPs)	+	+	+	reduction		[47]
*mdtG*	Multidrug efflux pump genes	Community wastewater collection point	+	+	+	reduction	China	[46]
*mdtH*			+	+	+			
*mdtN*			+	+	+			

na *—not analyzed; nd *—no difference; +—detected.

**Table 2 antibiotics-11-00747-t002:** Literature overview of different types of plasma discharges used for wastewater treatment.

Plasma Discharge	Discharge Characteristics	Strain	Antibiotic Resistance Profiles †	Volume	Initial Concentration	Matrices	Strain and Resistance Gene Reduction †	References
Discharge above liquid surface	V = 14 kV	*Staphylococcus aureus* (MRSA)	i-*mecA*		10^9^ CFU/mL	PBS * (pH = 7)	*Staphylococcus aureus* (MRSA) = 5 log	[27]
	Freq. * = 10 kHz						i-*mecA* = 0.8 log	
	Power = 2.94 W/cm^2^		e-*mecA*				e-*mecA* = 2.6 log	
	Time = 0 to 8 min	*E. coli* multi-resistant	i-*bla_TEM_*		10^9^ CFU/mL	PBS * (pH = 7)	*E. coli* multi-resistant = 3 log	[29]
			i-*tet*				i-*bla_TEM_* = 1.26 log	
			e-*bla_TEM_*				i-*tet* = 1.55 log	
			e-*tet*				e-*bla_TEM_* = 3.26 log	
							e-*tet* = 3.14 log	
Discharge in bubbles	V = 18 kV	*E. coli* multi-resistant	*tet C*	300 mL	10^8^ CFU/mL	PBS * (pH = 7)	*E. coli* multi-resistant = 7 log	[30]
	Freq. * = 50 Hz		*tet W*				*tet C* = 1.04 log	
	Power = 12 W		*bla_TEM-1_*				*tetW* = 0.61 log	
	Gas = Dry air at 2.5 L/min		*aac(3)-II*				*bla_TEM-1_* = 1.84 log	
	Time = 10 min		Integron gene *(intI1)*				*aac(3)-II* = 2.2 log	
							*intI1* = 2.3 log	
		*E. coli* multi-resistant	Integron gene *(intI1)*	500 mL	10^8^ CFU/mL	PBS * (pH = 7)	*E. coli* multi-resistant = 4.5 log	[31]
							*intI1* = 3.10 log	
Discharge in liquid	V = 500 V	*E. coli* multi-resistant	*tet A*	150 mL	10^8^ CFU/mL	Saline (0.9%)	*E. coli* multi-resistant = 7 log	[28]
	Current = 100 mA		*tet R*				*tet A* = 5.8 log	
	Power = 50 W		*aph A*				*tet R* = 5.4 log	
	Time = 30 min		Transposase gene *(tnpA)*				*aph A* = 5.5 log	
							*tnpA* = 5.5 log	

Freq. *—frequency; † i—intracellular gene; e—extracellular gene; PBS *—phosphate-buffered solution.

**Table 3 antibiotics-11-00747-t003:** ARG and ARB removal performance in the liquid phase by different AOPs.

Processes	Strain	Antibiotic Resistance Profiles	Volume	Removal Efficiency of ARB	Removal Efficiency of ARGs	Time	Energy Yield	Reference
Plasma	*E. coli*	*TetC, TetW, bla_TEM-1_, aac(3)-II,* and integron gene *(intI1)*	300 mL	7 log	1–2 log	10 min	18 kV	[30]
	*E. coli*	*tet A**,**tet R**,**aph A**,* and transposase gene *(tnpA)*	150 mL	7 log	5–6 log	30 min	500 V	[31]
Electrochemical	*E. coli*	*__*	150 mL	5 log	__	30 min	2.4 mA/cm^2^	[72]
Electrochemical oxidation/electro-Fenton	*E. coli*	*tetA*	300 mL	6 log	3–5 log	120 min	21.42 mA/cm^2^	[73]
Photo-Fenton/LED	*E. coli*	*bla_TEM-1_* and *tetA*	50 mL	6 log	6–8 log	30 min	19.2 mW/cm^2^	[18]
UV irradiation	*E.coli*	*tetA*	10 mL	4–5 log	3–4 log	1 min (ARB) and 30 min (ARGs)	20 mJ/cm^2^ (ARB) and 400 mJ/cm^2^ (ARGs)	[74]
UV/H_2_O_2_	*E. coli*	*amp^R^* and *kan^R^*	120 mL	5 log	3 log	5 min	100 mJ/cm^2^	[75]
Photocatalytic oxidation	*E. coli*	__	200 mL	3 log	__	120 min	80 W/m^2^	[19]

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
