# Peer review of "Degradation of Bacterial Antibiotic Resistance Genes during Exposure to Non-Thermal Atmospheric Pressure Plasma"

_antibiotics, 2022, doi:10.3390/antibiotics11060747_

Round 1

Author Response

Dear Reviewer,

Thank you for your remarks.

I have joint to this note the document our point-by-point answer to your comments.

Best regards,

Cristina Muja  

Reviewer 2 Report

The manuscript “Degradation of bacterial antibiotic resistance genes during exposure to non-thermal atmospheric pressure plasma” represents a substantial contribution to the literature with regards to the antibiotic resistance in hospital and household wastewater. The article is in line with Antibiotics journal topics. This type of review, apart from the official authorities’ reports, is lacking all over the world, and is therefore essential to picture the antibiotic resistant bacteria spread due to the incorrect and inefficient disinfection practices, whose the article clearly describes. Furthermore, the presented Tables are really useful to summarize the treatments and the relative doses and times. The article is well written and presented, the aim was centered. Small errors along the document have to be corrected. I suggest accepting the document for Antibiotics journal following minor revisions.

LINE 126: Correct “cfu/l” with “CFU/L”. Check the whole documents for similar corrections.

Author Response

Dear Reviewer,

Thank you for your remarks.

We made the modifications you asked.

Best wishes,

Cristina Muja 

Reviewer 3 Report

The authors described the degradation of bacterial antibiotic resistance genes during exposure to non-thermal atmospheric pressure plasma as a very interesting topic. The article is very well organized, well discussed, written in a proper way, and easy to read. The diagrams are very interesting. But before its final view, I would like to give some suggestions.

  1. LIne 53, authors mentioned that with the process of chlorine incomplete degradation of ARGs was observed. but there is another article that claims that Chlorination totally degrades the ARGs. kindly read that article and match both articles results. https://www.sciencedirect.com/science/article/abs/pii/S0045653515305385
  2. The authors have mentioned the role of plasma in degrading ARG, how about the nanoparticles? Can nanoparticles also control? 

Author Response

Dear Reviewer,

Thank you for your comments. 

Serveral modifications were made in the article to answer your remarks.

  1. Line 53, authors mentioned that with the process of chlorine incomplete degradation of ARGs was observed. but there is another article that claims that Chlorination totally degrades the ARGs. kindly read that article and match both articles results.

https://www.sciencedirect.com/science/article/abs/pii/S0045653515305385

The article was modified. We intoduced the reference you indicated (line 43).

We mentioned the results obtained by Zhuang et al (2015) [https://doi.org/10.1007/s11356-014-3919-z] concerning chlorine efficacy for ARGs degradation (line 57).

New text: 

"Zhuang et al investigated the inactivation sul1 and tetG, and showed that the exposure to 40 mg/L of chlorine for 60 minutes resulted in 1.65–2.28 log reduction of these genes. However, the chlorine concentration needed to obtain these results seems too high to be considered as treatment solution [17]."

  1. The authors have mentioned the role of plasma in degrading ARG, how about the nanoparticles? Can nanoparticles also control? 

Nanoparticle-based processes were mentioned in the article among the new processes for ARB and ARGs degradation (line 63) but a more detailed presentation of this type of treatment was not included in the article as the review objective was to present the potential of plasmas for wastewater treatment.

Best regards,

Cristina Muja